# Open-Winding Permanent Magnet Synchronous Generator for Renewable Energy—A Review

Abdur Rahman [1,*], Rukmi Dutta [1,*], Guoyu Chu [1], Dan Xiao [1], Vinay K. Thippiripati [2] and Muhammed F. Rahman [1]

[1] School of Electrical Engineering and Telecommunications, The University of New South Wales Sydney, Sydney 2052, Australia; g.chu@unsw.edu.au (G.C.)
[2] Department of Electrical Engineering, National Institute of Technology, Warangal 506004, India
* Correspondence: abdur.rahman@unsw.edu.au (A.R.); rukmi.dutta@unsw.edu.au (R.D.)

**Abstract:** The open-winding permanent magnet synchronous machines (OW-PMSMs) have recently been gaining more attention because of their fault-tolerant capability and power quality comparable to a 3-level converter-driven system. This paper reviews the common configurations of OW-PMSM when used as a generator, highlighting its shortcomings and benefits. The OW-PMSM with a common DC bus was found to be a promising direct-drive generator solution for wind energy conversion (WEC) systems considering fault tolerance, DC bus utilization, and power quality when appropriate control algorithms are in place. The presence of the zero-sequence current is the key disadvantage of the common DC bus configuration. The review highlights the algorithms that have been proposed to suppress the zero-sequence current of the OW-PMSM under healthy and various fault conditions, especially the open-circuit fault of semiconductor switch. Shutting down remotely located wind turbines because of faults, until they can be repaired, may not make economic sense. The OW-PMSM can offer the opportunity, to run a WEC system even under fault conditions albeit with low output power. This paper will assess the literature gaps in the existing control techniques that prevent the extension via a comprehensive review.

**Keywords:** open-winding permanent magnet synchronous machine; zero-sequence current; dual converter; model predictive control; direct torque control; space vector pulse width modulation

## 1. Introduction

The open-winding (OW) configuration for the electric machines was proposed for high-power AC drives to embed additional DC voltage sources and to provide fault tolerance [1–3]. The OW drives were also proposed as an alternative to modular multi-level converters for improving power quality [4]. The application of open-winding topologies is also gaining attention in distributed generations. In a microgrid with renewable energy sources, an additional converter is often required to set up a DC link voltage as shown in Figure 1. Such systems can benefit from the use of an open-winding (OW) generator [5].

Due to high power density and efficiency, the permanent magnet synchronous generator (PMSG) is preferred in direct-drive wind energy conversion (WEC) systems. They are also widely used as engine-driven portable generators in distributed generation and microgrids. In general, back-back full power-rated converters are necessary as the interface between the PMSG and grid or local load. The OW configuration of a direct-drive PMSG can lower the VA rating of individual switches of the converters when half-controlled converters were used [6]. Despite the advantages of fault tolerance and power quality, the use of OW-PMSG in a microgrid, distributed generation, and WEC is limited due to several challenges.

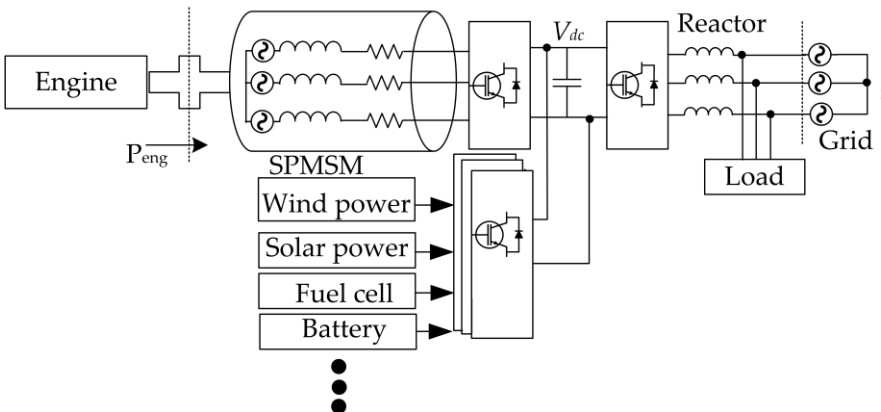

**Figure 1.** Microgrid with multiple sources.

In an OW configuration, the neutral connection of the machine is opened, and each side of a phase winding is connected to one leg of a two-level voltage source converter (VSC). Figure 2 shows the OW configuration for a 3-phase AC machine. The converter of each side could be connected to two separate DC sources or a single common DC source. In Figure 2, $V_{dc1}$ and $V_{dc2}$ represent two isolated DC buses.

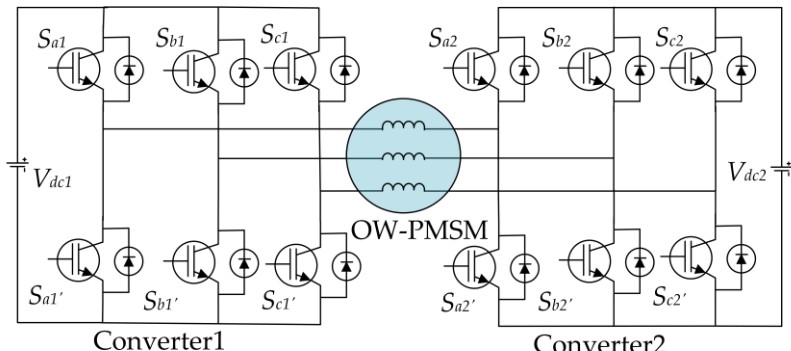

**Figure 2.** Open-winding configuration of a 3-phase AC machine (isolated DC bus configuration).

The advantages of OW configuration over conventional Y-connected PMSM include multi-level operation, reduced converter capacity, more control flexibility due to a higher number of voltage vectors, and enhanced fault tolerance with the freedom to control each phase [7,8]. Compared to a 3-level multi-level converter, a dual converter system of OW configuration requires fewer capacitors than the flying capacitor multi-level converters, fewer isolated DC sources than H-bridge converters, and fewer diodes than the Neutral Point Clamped multi-level converters [9]. However, if two separate DC sources are used in OW configurations, it increases the size of the drive system. Contrary to this, if a single DC source as a common DC bus between the two converters is used, zero-sequence current circulates in the systems leading to additional torque ripples and losses in the electric machine.

Various control strategies such as proportional integral/proportional resonant (PI/PR)-based vector control, model predictive control (MPC), and direct torque control (DTC) have been proposed for OW-PMSG with a common DC bus [10–24]. A large part of the control literature related to the OW topologies is devoted to the reduction of the zero-sequence current in the common DC bus configuration [10–24]. A comprehensive review of these strategies allows qualitative assessment of the performance of OW topologies. In [8], the OW drive topologies were reviewed considering the DC bus voltage utilization, output power quality, and fault-tolerant capability for aerospace applications. However, a comparative study among the existing control schemes is yet to be conducted. Moreover,

the requirement of the aerospace application is significantly different from the PMSG-based WEC system. This paper presents a comprehensive review of the OW-PMSG with the view of identifying research challenges and future direction when applied as a direct-drive WEC system. It should be mentioned that many control schemes reviewed in the paper are equally applicable to both the OW-PMSM and OW-PMSG. Hence, the word OW-PMSM is adopted for convenience.

The paper is organized into seven sections. After the introduction in Section 1, Section 2 describes the most common configurations of the OW-PMSM and their advantages and shortcomings when applied in a generation system. Section 3 discusses the model, equivalent circuit, and the common switching schemes of OW-PMSM. Section 4 reviews various control schemes proposed for OW-PMSM with a common DC bus. Section 5 compares the performance of OW-PMSM for the existing various control strategies, and a discussion on challenges and future direction for the OW-PMSM when used as a renewable energy generation system has been presented in Section 6. Finally, Section 7 concludes the paper.

## 2. OW-PMSM Configurations

Based on the DC bus configuration and type of converters, the OW-PMSM systems can be classified broadly as—(A) with isolated two DC buses, (B) with a common DC bus, and (C) with a floating capacitor and a DC bus [25]. If the bidirectional flow is not required, half-controlled converters can be used as an alternative, which reduces device VA ratings [6,26]. Each configuration is described below highlighting the advantages and disadvantages when used with OW-PMSM.

### 2.1. Isolated Two DC Buses

In the configuration of Figure 2, both the converters are connected to two isolated DC buses allowing maximization of DC voltages without any common mode reactance. The key advantage of this topology is no circulation of zero-sequence current (ZSC). The fault tolerance capability is also enhanced since the machine can operate with one healthy converter similar to a single converter system albeit with half output capability as the available DC bus decreases [25]. The isolated two DC bus OW configuration was first proposed with two identical DC buses (or sources) [1,2]. However, later, it was found that the two DC buses (or sources) could have different values and the configuration can behave like a 3-level dual input converter [27,28]. It was also shown in [28] that power between the two converters can be shared unequally. In motoring operation, the two buses could be supplied from two electrically isolated DC sources. In a microgrid application with distributed generation, one of the DC buses can be connected to a battery or a photovoltaic source, whereas the other DC bus can be connected to a DC (or AC) grid. In the case of an AC grid, an additional converter will be required.

A direct grid connection of one side of the OW of a PMSG was proposed in [5]. When the generator is not operating, the winding inductance can serve as the filter reactor for the DC grid connected back to the AC grid via the DC-AC converter. One of the key disadvantages of such an OW system is that fault tolerance capability reduces. In addition, the copper loss of the generator contributes to the total loss even when the generator is not running.

In general, the overall size of the isolated DC bus OW topology is much larger than other comparable converter-driven systems including the OW common DC bus configuration. Hence, this configuration is recommended mostly for those applications that benefit significantly from two isolated DC sources in which size increase can be overlooked.

### 2.2. Common DC Bus

The OW-PMSM system supplied from a common DC bus (or source) is shown in Figure 3. The primary benefits of this topology are having a simplified system structure and reduced system size and cost. However, the common DC bus creates a zero-sequence path

due to the direct connection between the two converters allowing the zero-sequence current (ZSC) to flow [10]. The flow of ZSC reduces the system efficiency by increasing the total harmonic distortion (THD) of phase current and more importantly, it increases the system loss and torque ripple [29]. The torque ripple is important for a WEC system because of its low speed, and it can cause wear and tear in the turbine shaft. The sources of ZSC may be attributed to the machine's third harmonic back emf, converter nonlinearity, and the PWM modulation itself [30]. In addition, the cross-coupling inductances between the *dq*-axis and zero axis also contribute to the ZSC, especially for fractional-slot-concentrated winding PMSM [30]. Many studies have been performed, as will be discussed in Section 4, to suppress the ZSC to improve the steady state and dynamic response of the machine in terms of reduced phase current THD and torque ripples.

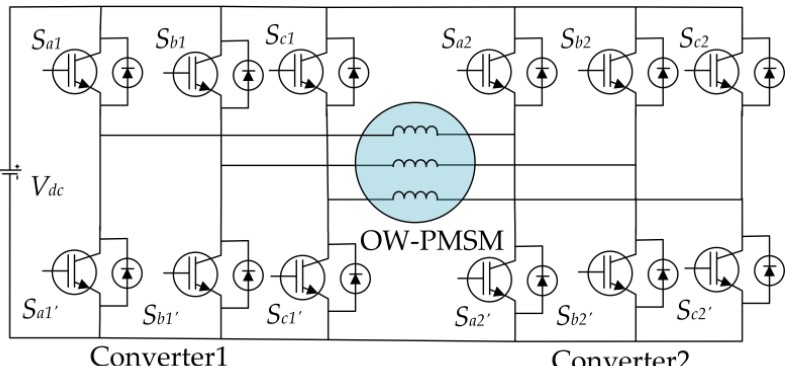

**Figure 3.** OW-PMSM configuration with common DC bus.

In terms of fault tolerance, a comprehensive study in [31,32] showed the effectiveness of this configuration. It was demonstrated in [31] that the output torque can be maintained at 70% of rated torque for switch/diode faults associated with the same phase, whereas for multi-phase open circuit faults associated with a maximum of six switch/diode faults the output torque can still be maintained at 57% of its rated value. The fault-tolerant capability of this topology was further enhanced for up to three-leg open circuit faults in [32] without compromising the output capacity by utilizing six bidirectional solid-state relays. Another advantage is the better utilization of the DC bus compared to the isolated DC bus topology. According to [7,8], the maximum DC bus utilization factor of the common DC bus configuration is two. However, this factor may reduce due to the suppression of ZSV to minimize ZSC. For the same reason, this topology may degrade from multi-level to two-level operations [8]. The power quality of an OW-PMSM is compromised if ZSC is not suppressed effectively [8].

### 2.3. Floating Capacitor

The general structure of an OW topology with a floating capacitor is shown in Figure 4, where $V_{dc}$ and $V_{FC}$ represent the DC bus and floating capacitor voltage, respectively. Compared to the common and isolated DC bus topology, this configuration neither allows zero sequence current to flow nor requires any isolation for the supply. However, the main challenge lies in maintaining the floating capacitor voltage [33]. In addition, the performance can degrade while attempting to control the floating capacitor voltage and maintain multi-level operation simultaneously [34,35]. Although this topology shows superior performance in terms of DC bus voltage utilization and power quality, the fault-tolerant capability becomes limited [8]. The main reason for the reduced fault tolerance is that the system can maintain performance only if the fault occurs on the floating capacitor side [36,37]. When a switch or diode fails, for example, $S_{a2}$ in Figure 4, the corresponding $S_{b2}$ and $S_{c2}$ are turned off, while the lower three switches are turned on to make a neutral point allowing the machine to operate like a conventional Y-connected PMSM with

reduced output power [34]. The fault-tolerant capability of this topology was enhanced by introducing three solid-state relays in [38].

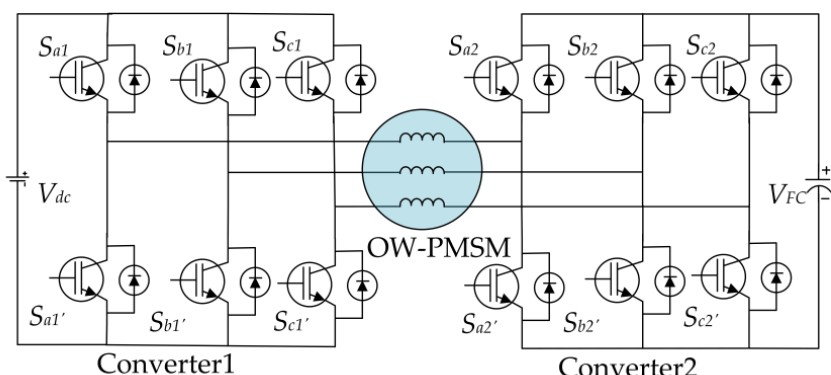

**Figure 4.** OW-PMSM configuration with a floating bridge capacitor.

### 2.4. Semi-Controlled

The OW-PMSM fed by a semi-controlled converter for both common and isolated DC bus configurations is shown in Figure 5. In this figure, VSC represents the voltage source converter. This topology integrates a voltage source converter and diode bridge rectifier, thus reducing the system cost and improving the system's reliability while the driving circuit becomes simplified [39]. However, the modulation range of the voltage source converter reduces, due to the diode rectifier voltage output becoming current polarity dependent [39]. Moreover, the DC bus voltage utilization ratio reduces to 0.866 for the common DC bus topology compared to the isolated one, due to the presence of zero-sequence current [40].

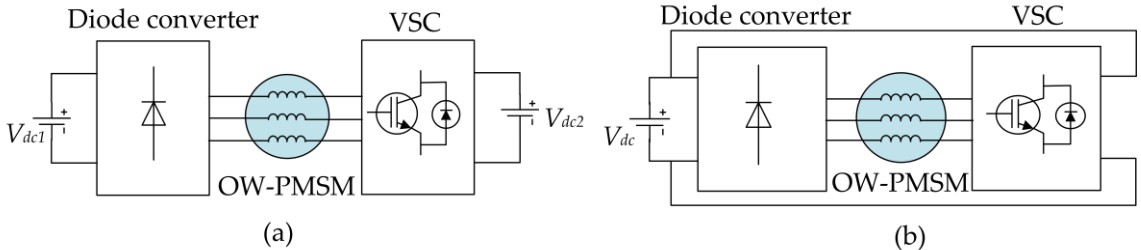

**Figure 5.** Semi-controlled OW-PMSM configuration. (**a**) Isolated DC bus. (**b**) Common DC bus.

## 3. OW-PMSM with Common DC Bus

### 3.1. OW-PMSM Model

The phase voltage equations of OW-PMSM can be expressed as [41]

$$
\begin{bmatrix} u_a \\ u_b \\ u_c \end{bmatrix} = \frac{d}{dt} \begin{bmatrix} \psi_a \\ \psi_b \\ \psi_c \end{bmatrix} - \begin{bmatrix} R_s & 0 & 0 \\ 0 & R_s & 0 \\ 0 & 0 & R_s \end{bmatrix} \begin{bmatrix} i_a \\ i_b \\ i_c \end{bmatrix} \tag{1}
$$

where $u_{abc}$, $R_s$, $i_{abc}$, and $\psi_{abc}$ represent the phase voltage, resistance, current, and stator flux, respectively. The stator flux can be written as [10]

$$
\begin{bmatrix} \psi_a \\ \psi_b \\ \psi_c \end{bmatrix} = - \begin{bmatrix} L_s & M_s & M_s \\ M_s & L_s & M_s \\ M_s & M_s & L_s \end{bmatrix} \begin{bmatrix} i_a \\ i_b \\ i_c \end{bmatrix} + \begin{bmatrix} \psi_r \cos(\theta_r) + \psi_{3r} \cos(3\theta_r) \\ \psi_r \cos(\theta_r - \frac{2\pi}{3}) + \psi_{3r} \cos(3\theta_r) \\ \psi_r \cos(\theta_r + \frac{2\pi}{3}) + \psi_{3r} \cos(3\theta_r) \end{bmatrix} \tag{2}
$$

here, $L_s$ and $M_s$ are self and mutual inductances, $\psi_r$ and $\psi_{3r}$ are rotor fundamental and third harmonic flux linkage, respectively, and $\theta_r$ is rotor electrical position. In an OW-PMSM, the presence of the third-order harmonic in the rotor flux linkage is a major source of the ZSC.

Assuming the rotor flux is aligned with the *d*-axis, the stator flux of *abc* frame, when transformed into the rotor *dq0* reference frame, can be shown as

$$
\begin{bmatrix} \psi_d \\ \psi_q \\ \psi_0 \end{bmatrix} = T_{\frac{3S}{3R}} \begin{bmatrix} \psi_a \\ \psi_b \\ \psi_c \end{bmatrix}
$$
$$
= - \begin{bmatrix} L_d & & \\ & L_q & \\ & & L_0 \end{bmatrix} \begin{bmatrix} i_d \\ i_q \\ i_0 \end{bmatrix} + \begin{bmatrix} \psi_r \\ 0 \\ \psi_{3r}\cos(3\theta_r) \end{bmatrix} \tag{3}
$$

here, $L_d$, $L_q$, $L_0$ are *d*, *q*, and *0*-axis inductance respectively; $i_d$, $i_q$, $i_0$ are *d*, *q* and *0*-axis component of phase current.

The voltages in the *dq0* reference frame can be expressed as,

$$
\begin{bmatrix} u_d \\ u_q \\ u_0 \end{bmatrix} = \begin{bmatrix} -L_d\frac{di_d}{dt} - Ri_d + \omega L_q i_q \\ -L_q\frac{di_q}{dt} - Ri_q - \omega L_d i_d + \omega\psi_r \\ -L_0\frac{di_0}{dt} - Ri_0 - 3\omega\psi_{3r}\sin(3\theta_r) \end{bmatrix} \tag{4}
$$

Similarly, the electromagnetic torque can be expressed as:

$$
T_e = \frac{3}{2}n_p[L_q i_d i_q + (\psi_r - L_d i_d)i_q - 6\psi_{3r}i_0\sin(3\theta_r)] \tag{5}
$$

where $n_p$ is number of pole pairs. It can be seen from (5) that ZSC contributes to a ripple component in the induced torque, and this is one of the reasons why ZSC suppression in an OW-PMSM is essential.

### 3.2. Zero-Sequence Equivalent Circuit

Both the machine and the converters contribute to the circulation of ZSC in an OW-PMSM. The representation of the sources of ZSC in an equivalent circuit is proposed in [30]. The major source of ZSC is the third harmonic back emf in the machine side [10,30]. In addition, it was shown in [30] that a cross-coupling effect between the *dq*-axis and zero-axis that exist in fractional-slot-concentrated winding (FSCW) OW-PMSM also contributes to ZSC generation from the machine side. The cross-coupling effect exists because, in the FSCW PMSM, the mutual inductance is much smaller than the self-inductance. As a result, the cross-coupling inductances are not zero in the *dq* inductance matrix [30]. On the converter side, the nonlinearity due to the dead time effect contributes to the zero-sequence voltage and current [30,42]. The other major source of ZSV is the modulation strategy of the dual converter.

The equivalent zero-sequence circuit with the ZSV sources is shown in Figure 6. The ZSV source due to cross-coupling of inductance can be represented as a current-dependent voltage source whereas the ZSV, due to the dead-time effect, is a function of phase currents [30]. Combining all four sources, the zero-sequence differential equation was derived in [30] as,

$$
\begin{aligned}
u_0 &= u_{0-dt} + u_{0-id} + u_{0-iq} + e_0 + R_s i_0 + L_0\frac{di_0}{dt} \\
&= \frac{f(i_a)+f(i_b)+f(i_c)}{3} - 3\omega i_d L_\Delta\sin(3\omega t) - 3\omega i_q L_\Delta\cos(3\omega t) - 3\omega\psi_{3r}\sin(3\omega t) + R_s i_0 + L_0\frac{di_0}{dt}
\end{aligned} \tag{6}
$$

where $u_0$ represents modulated ZSV, $u_{0-dt}$ is the ZSV due to the dead-time effect, $u_{0-id}$ and $u_{0-iq}$ arise from cross-coupling between the $dq$ and zero axis, and $e_0$ is the third harmonic back emf voltage. The phase distortion voltage error $f(i_x)$ is given in [43] as below,

$$f(i_x) = 2\Delta U \left( \frac{1}{1 + e^{-k\,i_x}} - \frac{1}{2} \right) \tag{7}$$

where $i_x$ is an arbitrary phase current, $\Delta U$, and $k$ are distortion voltage model parameters.

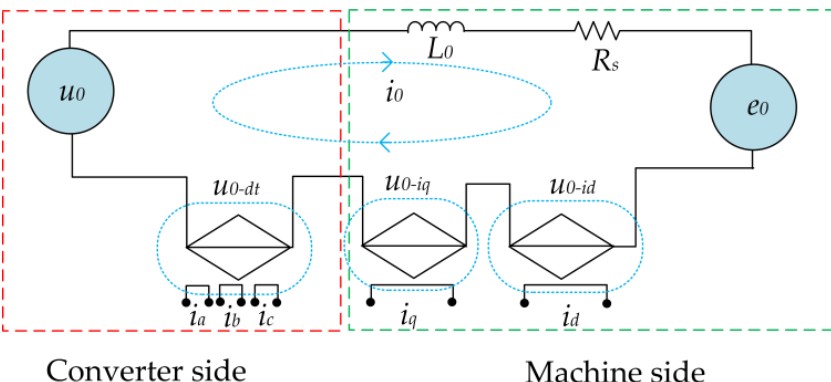

**Figure 6.** Zero-sequence equivalent circuit of the OW-PMSM system.

### 3.3. PWM Schemes for Eliminating ZSC

The output of a 2-level voltage source converter (2-L VSC) can be represented by eight space vectors. The eight space vectors of each converter can be represented as a hexagon in the stationary α-β plane as shown in Figure 7 in which VSC1 modulates vectors from $V_0$ to $V_7$ and VSC2 modulates vectors from $V_0{}'$ to $V_7{}'$. The switching states of each voltage vector are given in brackets, where 1 represents the corresponding upper switch device is on and 0 means the corresponding lower switch device is on. Since each converter can assume eight independent states, the dual-converter scheme can produce a total of 64 space-vector combinations distributed over 19 space-vector locations (A, B, ... S) as shown in Figure 8, where $ij$ ($i, j = 0, 1, \ldots 7$) denote the synthesized VV of $V_i$ and $V_j{}'$. It can be observed from Figure 8 that the switching states of dual converter configuration can lead to seven different levels of zero-sequence voltage, i.e., 0, $\pm 1/3\ V_{dc}$, $\pm 2/3\ V_{dc}$, $\pm V_{dc}$.

It is worth mentioning that initially, research related to OW focused on developing the optimum switching schemes for dual converters [12,13,44–50]. The modulation scheme can be divided into two categories-coupled and decoupled based on whether the dual converter is modulated as a single or two separate systems. In the coupled strategy, the reference voltage vector is modulated as a single system, whereas for the decoupled method, the reference vector is divided into two, and each part is synthesized by the individual converter. For example, the alternate sub-hexagonal center (ASHC) PWM strategy proposed in [44,50–52] is a coupled modulation strategy. In the ASHC shown in Figure 9a, the dual converter can be treated as a single multi-level converter. In this strategy, each converter is clamped alternately when the other converter is switched to synthesize the reference voltage vector. When the reference vector OT is in sector 1, converter 1 can be clamped to switching state 1 (100) to produce OA, and the component AT can be realized by switching the second converter. Alternately, the component OA can be realized by clamping converter 2 to state 4′ (011), and AT can be realized by converter 1. The process repeats in each sector. Hence, in this method, the loss is equally divided between the two converters [52]. The benefit of this strategy is that multi-level output can be observed at all times [8]. On the other hand, in the decoupled PWM strategy [10,45,53,54], the reference voltage vector is

divided between the two converters as shown in Figure 9b and realized by modulating both converters simultaneously. From Figure 9b, the reference vector can be written as

$$U_{ref}e^{j\theta} = u_1 e^{j\theta_1} - u_2 e^{j\theta_2} \tag{8}$$

where, $U_{ref}$ is the reference vector, $u_1$ and $u_2$ are the voltage vectors from converters 1 and 2, respectively, $\theta$, $\theta_1$, and $\theta_2$ are the phase angle corresponding to $U_{ref}$, $u_1$, and $u_2$, respectively. From (8) and Figure 9b, it can be seen that the decoupling strategy has four user-defined variables, resulting in greater control flexibility. Although this control scheme offers several advantages such as achieving multi-control objectives and improved fault tolerance, the resultant load voltage may be unpredictable [55].

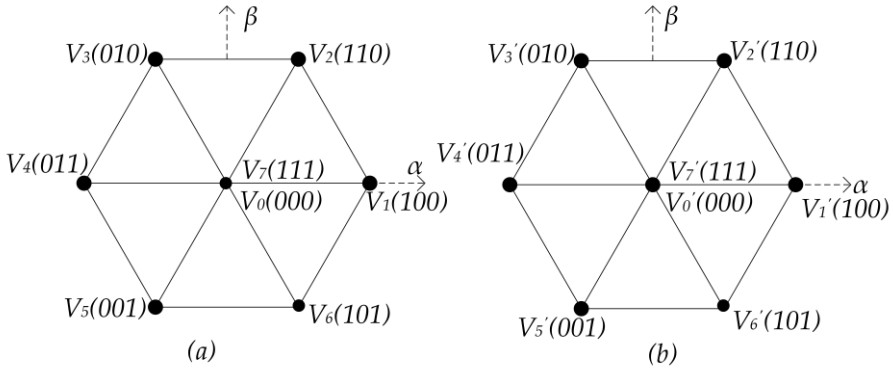

**Figure 7.** Space vectors from converter 1 (**a**) and converter 2 (**b**).

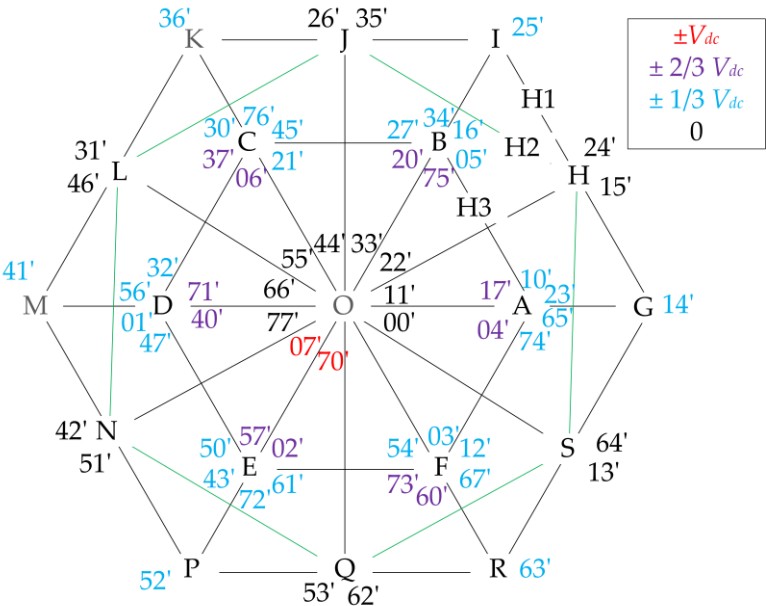

**Figure 8.** Space vector combinations for the dual converter scheme; letters A, B, . . . F indicate the space vector locations for individual converter; H, J, L, . . . S refer to the space vector locations for zero ZSV; G, I, K, ...R refer to the highest possible space vectors for dual converter.

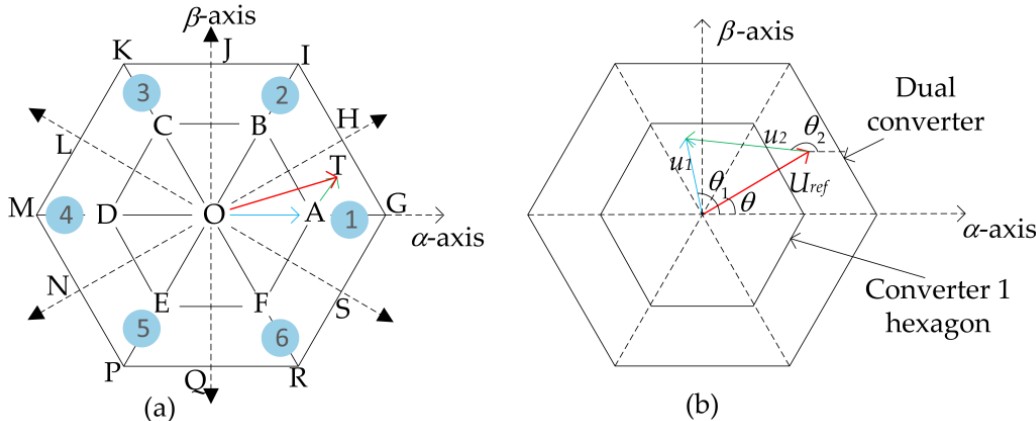

**Figure 9.** PWM switching schemes. (**a**) Principle of alternate sub-hexagonal center PWM switching strategy; 1,2, . . . 6 denote the sector numbers; meaning of letters A, B, ...S has been explained in Figure 8. (**b**) Principle of decoupled switching scheme.

In [12,13], the switching states that produce zero ZSV were employed in the open-winding induction motor (OW-IM) drive. When such strategies were used, the number of space vectors was reduced to 20 (from 64), resulting in a reduced fundamental component of the output voltage and underutilization of the DC bus. Consequently, boosting the DC bus voltage is required to obtain the rated voltage at the machine terminal. Moreover, these schemes did not consider the dead-time effect and device voltage drops. Hence, the complete elimination of ZSC was not possible with these switching schemes even in OW induction machines. The common mode chokes and pulse-based dead-time compensation (PBDTC) strategies were implemented in [56] along with the SVPWM scheme that does not contribute to ZSV. However, such methods need further consideration in an OW-PMSM, as the machine side sources of ZSC must also be considered along with the converter switching and non-linearity.

## 4. Control of OW-PMSM with a Common DC Bus

After the switching scheme, the applied control algorithms play a significant role in the performance of a machine drive. This section reviews the existing control strategies of the OW-PMSM with a common DC bus configuration. Although most of the existing control strategies in the literature have been discussed for motors, the same strategies can still be applied to generators with the relevant modification to the reference signals. The commonly used control schemes such as the rotor field-oriented control (RFOC), direct torque control, and model predictive control were implemented for the OW-PMSM drive in addition to a ZSC suppression scheme. Many other newly developed non-linear control schemes for PMSM are yet to be investigated for OW-PMSM. It is noteworthy that non-linear control schemes such as MPC offer ZSC suppression without any additional controllers and hence, in recent years, the MPC has been investigated vigorously for OW-PMSM [21,23,24,57–62].

PI-based FOC control of $i_d$ and $i_q$ current are very well documented and will not be elaborated in this paper. However, when the FOC scheme is adopted for an OW-PMSM, the ZSC must be controlled with an additional controller. The following section first discusses ZSC suppression in the FOC scheme, which is then extended to the DTC and MPC.

### 4.1. FOC-Based SVPWM Control

The block diagram for this control strategy is shown in Figure 10. In this strategy, the reference $d$ and $q$ voltages are synthesized from the two proportional-integral (PI) controllers. In addition to these, the reference zero-sequence voltage is obtained from a ZSC controller. The PI, proportional resonant (PR), and hysteresis controllers were proposed as the additional ZSC controller for FOC-based SVPWM [10,11,63]. Several ZSC suppression strategies have been proposed in the existing literature. It was shown in [44,45] that

the placement of the effective time block within a PWM period significantly affects the ZSV distribution, and with a suitable offset, ZSV can be eliminated in the average sense over a PWM period to suppress ZSC. However, the ZSV suppression capability of the proposed methods may be reduced for the high voltage range. In the discontinuous PWM schemes, the effective time block can be placed at either side of a switching period. It allows switching loss reduction by 33%, although ZSV increases considerably. Hence, it is not suitable for an OW dual converter system. However, an alternative to these, the zero-vector redistribution can be used to synthesize ZSV that can suppress ZSC. A zero-vector redistribution PWM strategy presented in [10] suppressed the ZSC and enhanced the modulation range simultaneously. In this method, the duration of zero vectors (0 0 0) and (1 1 1) were determined based on the ZSV produced by active vectors. However, the method considered only the third harmonic flux linkage as the source of the ZSC from the machine side. Moreover, the proposed scheme requires 12 switching actions within one PWM period, resulting in a relatively high switching loss. In addition, the inclusion of a zero vector (1 1 1) in the modulation process can increase the shaft leakage voltage that can lead to a bearing fault [64,65]. An improved modulation strategy proposed in [14] used active voltage vectors to produce the reference ZSV for suppressing ZSC. The issues of high switching loss and shaft leakage voltage were eliminated since the zero vector (1 1 1) was not used.

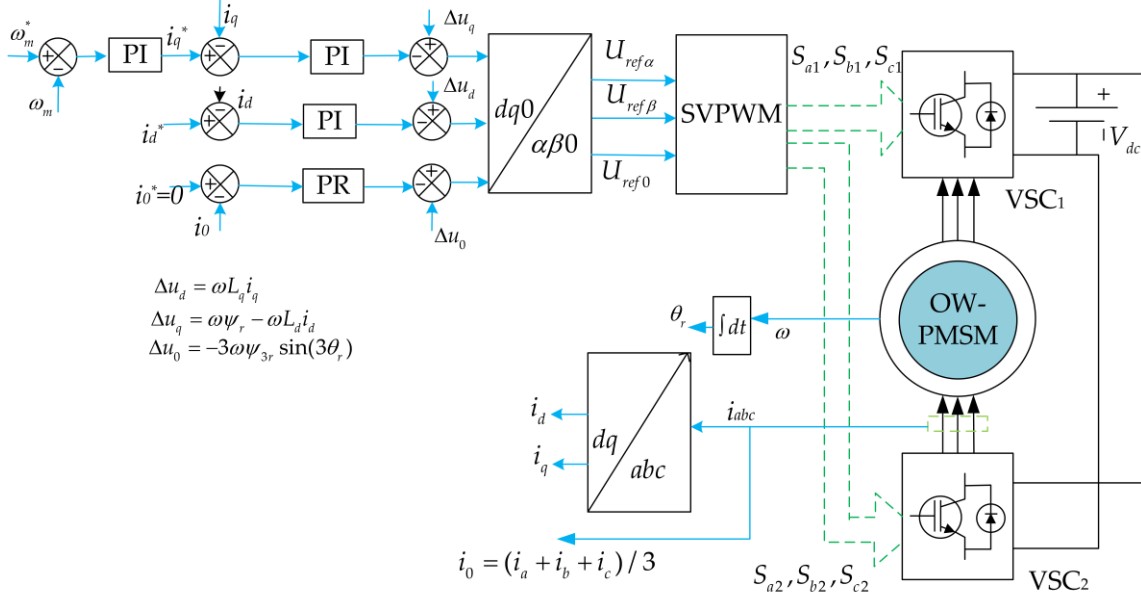

**Figure 10.** Block diagram for SVPWM control of OW-PMSM; "*" indicates a reference value; $U_{ref\alpha}$, $U_{ref\beta}$, $U_{ref0}$ are $\alpha$, $\beta$ and $0$-axis component of reference phase voltage.

It was shown in [30] that the effect of ZSV due to the converter non-linearity becomes significant when the phase current moves away from the linear region of phase voltage error (when expressed as a sigmoid function). To neutralize the resultant ZSV caused by different ZSV sources including converter non-linearity, a frequency adaptive PR controller has been adopted to synthesize the required ZSV in [30]. The PR controller improves the steady-state performance considerably. However, the suppression of ZSC during the transient conditions was not as good as that of the steady state. The PI/PR-based ZSC controller often involves complex parameter tuning, and the ZSC suppression can be affected in the high modulation index range. A hysteresis controller for ZSC was used in [11] to avoid overmodulation of ZSV at a high fundamental modulation index. The hysteresis controller is also more amenable to parameter uncertainty in the machine model. The proposed technique has a faster dynamic response and simpler control process compared to [10]; however, the steady-state performance declines and the linear modulation range

also decreases marginally. The reference [66] further investigated hysteresis control based SVPWM to extend the linear modulation range by 12.7%.

From the perspective of torque ripple suppression, a novel current injection-based torque ripple suppression strategy with reduced switching frequency was introduced in [54], in which the torque ripple produced by the injected current and ZSC cancel out each other. However, the ZSC was not suppressed in the proposed control scheme, and the effective modulation range is decreased to 0.78 as the injected current disturbance was incorporated within the modulation range. Considering non-sinusoidal back-EMFs and ZSC, a torque ripple minimization technique by combining PI controller with back emf feedforward compensation was proposed in [67].

For high-speed machines, where the phase inductance and consequently the zero-sequence impedance (ZSI) are very small, the small time constant of the zero-sequence circuit can produce high-frequency and high-intensity current ripple, which might be harmful to the switching devices [15]. Therefore, the ZSC suppression strategies discussed in [10,14,30,66] cannot be used for such machines with low ZSI. To suppress the ZSC for high-speed machines, the reference [15] presented a hybrid modulation scheme where one converter (IGBT switch based) was clamped to work in square wave mode and the other converter (SiC switch based) operated with SVPWM to suppress the ZSV instantaneously. The proposed algorithm can effectively eliminate the ZSV from the converter side only. The ZSC due to the machine's third harmonic back emf, cross-coupling voltages, and converter dead time were not eliminated. Elimination of the effect of third harmonic back emf using the dead-time-generated phase distortion voltage was attempted in [42]. Here, the pulse-based deadtime compensation (PBDTC) proposed in [15] was combined with the modulation and a DT (dead time) hysteresis controller. However, the ZSC controller required an FPGA implementation due to the high dynamics of the ZSC.

The presented control strategies in [10,14,15,30,66] did not consider shaft leakage voltage due to the common mode voltage (CMV) that may adversely affect the motor bearings. To simultaneously control the ZSC and CMV of the OW-PMSM, a modulation technique was proposed in [29], which adjusts the duration of each phase voltage. In addition, the device's dead-time effect on CMV control was also investigated to verify the robustness of the proposed algorithm. Shen et al. [53] introduced a phase shift-based sinusoidal PWM (PS-SPWM) scheme for OW-PMSM, where the location of voltage pulses is adjusted to suppress the ZSC. The proposed strategy also reduced the motor vibration and acoustic noise in addition to the small high-frequency current ripple compared to the conventional SVPWM scheme by maintaining the double-frequency effect that can eliminate odd switching harmonics from phase current.

Refs. [16–18] presented a sensorless control strategy for OW-PMSM where the rotor position was estimated from the reconstructed third harmonic back electromotive force by deploying ZSC. The position estimation is sensitive to zero-sequence circuit parameters. Ref. [16] investigated that the effect of phase resistance mismatch on position estimation becomes significant in the low-speed region, whereas the estimation error is more sensitive to zero-sequence inductance deviation in the high-speed range.

### 4.2. Direct Torque Control

The block diagram of direct torque control (DTC) for OW-PMSM with a common DC bus is shown in Figure 11. In this method, torque and flux are taken as control objectives, and the optimal voltage vector is selected based on a look-up table. The electromagnetic torque of OW-PMSM is directly controlled by voltage vectors, thus no tedious weighting factor tuning is necessary here [68]. Moreover, due to the elimination of the inner current control loop, the dynamic response is significantly enhanced [19]. A DTC based on duty ratio modulation for zero-sequence current suppression has been proposed in [19], where the space vectors that produce zero ZSV were employed for the torque and flux control, and zero voltage vectors (VVs) (0 0 0) and (1 1 1) were used to synthesize the required ZSV to compensate the third harmonic back emf. In [19], a deadbeat zero-sequence current

controller is used to suppress the ZSC, and the torque and stator flux ripple were reduced considerably. However, the proposed system adopts a complex duty cycle optimization to enhance the system performance. Moreover, the introduction of zero VVs reduced the usable voltage of the synthetic voltage vectors and the modulation range while the switching frequency increased. To decrease the switching loss and suppress the ZSC effectively, a modified look-up table-based DTC was proposed in [20], where two VVs were applied at each switching period, one for controlling the torque and flux, and the other one for producing the reference zero-sequence voltage. The proposed strategy maintains the advantages of low computational complexity, simple structure, and enhanced dynamic performance of conventional DTC by modulating only one switching phase in each control period. However, the problem associated with the hysteresis controller was not addressed, and the phase current THD was as high as 16.3%.

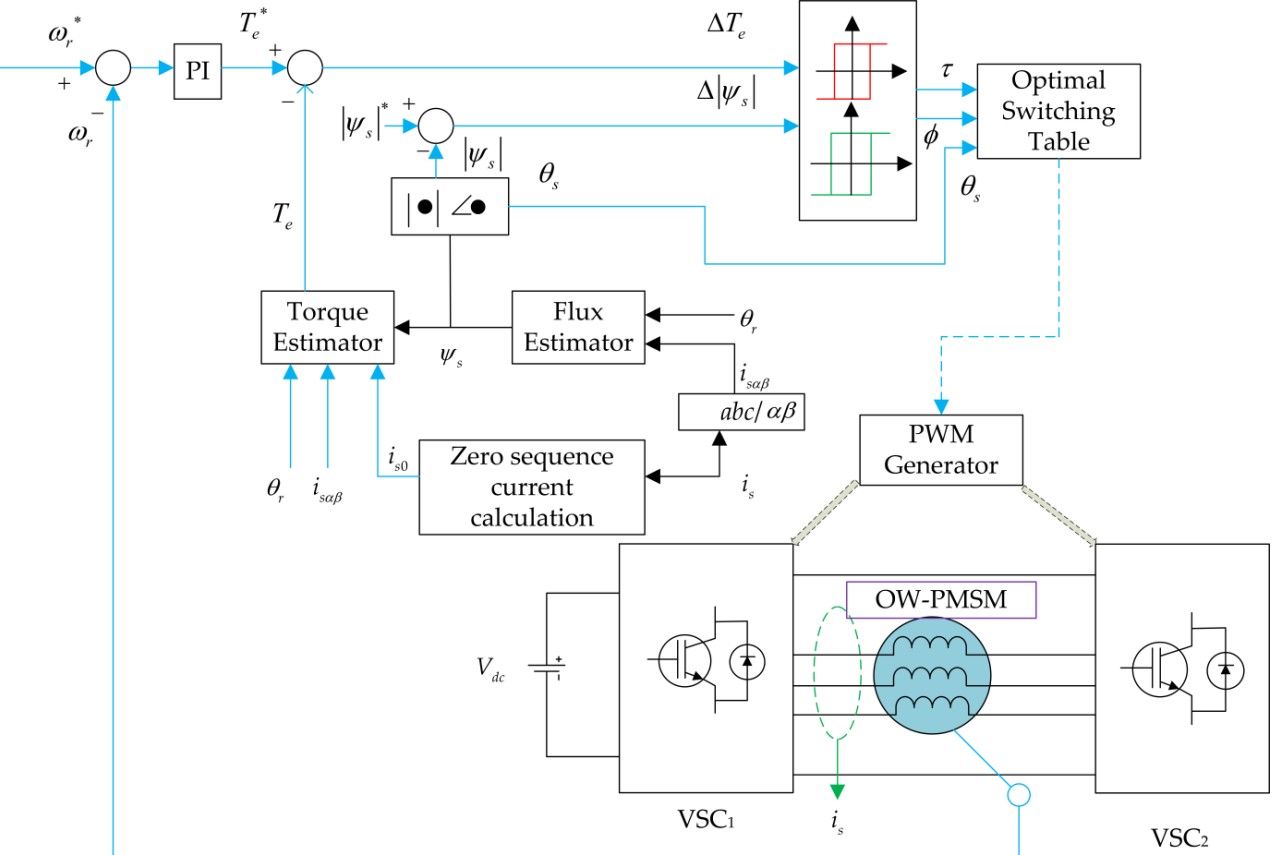

**Figure 11.** Control diagram of the DTC for OW-PMSM with common DC bus; "*" indicates a reference value; $\theta_r$, $\omega_r$, $T_e$, $\psi_s$ are rotor position, speed, electromagnetic torque and stator flux respectively; $i_s$, $i_{s\alpha\beta}$, $i_{s0}$ are phase current, $\alpha\beta$ component of phase current and zero-sequence current respectively.

To suppress the torque ripple using DTC, a modified voltage switching scheme was presented in [69], in which two different look-up tables were established based on the speed of the drive system to reduce the torque and flux ripple. In [70], further analyses were conducted to modify the switching table and incorporate additional voltage vectors to reduce the torque and flux ripple. However, the effect of ZSC on torque was not investigated in [69,70].

### 4.3. MPC

Several different predictive controllers have been proposed in the literature such as deadbeat predictive control, hysteresis-based predictive control, finite control set MPC (FCS-MPC), and continuous control set MPC (CCS-MPC) [71]. The FCS-MPC is the most

promising one among these different control schemes as it uses the switch states and directly employs the optimal control action [71,72]. The block diagram of the conventional model predictive current control is shown in Figure 12. The currents $i_{dq0}$ can be predicted at $(k+1)^{th}$ instant from the measured currents to compensate for the hardware delay. The next step is to predict the currents at $(k+2)^{th}$ instant using the available voltage vectors (VVs) and calculate a cost function. The switching state corresponding to the VV that minimizes the cost function is directly applied to the converter in the upcoming switching period.

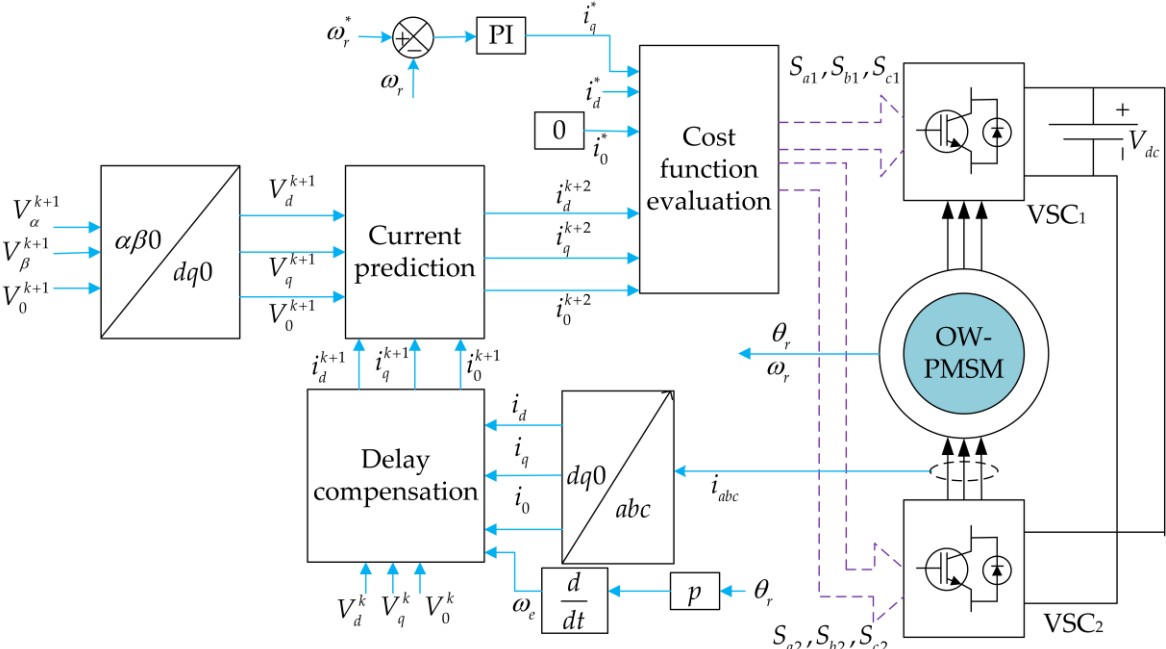

**Figure 12.** Schematic of the finite control set-MPCC for OW-PMSM; "*" means a reference value; $V_d{}^k$, $V_q{}^k$, $V_0{}^k$ are $d$, $q$ and 0-axis component of phase voltage at $k^{th}$ instant; $V_\alpha{}^{k+1}$, $V_\beta{}^{k+1}$, $V_0{}^{k+1}$ are $\alpha$, $\beta$, and *0*-axis component of phase voltage at $(k+1)^{th}$ sampling period; $p$ is number of pole pairs.

For the OW-PMSM system, many MPC control strategies have been proposed. They can be divided mainly into two categories—model predictive torque control (MPTC) and model predictive current control (MPCC). The control constraints for MPTC are generally electromagnetic torque and stator flux. Furthermore, for the OW-PMSM system, the zero-sequence current needs to be accounted into the cost function to suppress the ZSC. All these variables have different units, so weighting factors need to be calculated, which requires extensive experimentation [21]. To avoid the weighting factor calculation, one multi-objective cost function can be divided into several single objective cost functions, and the optimal vector that minimizes the average ranking of all cost functions could be selected [73,74]. Another way to avoid the weighting factor is to convert the multi-variable cost function into its equivalent single-variable cost function [21]. The control constraints were converted into equivalent torque [75] by transforming stator flux into equivalent reactive torque, although the ZSC control variable was ignored in the cost function. The weighting factors are also eliminated by transforming the control objectives into equivalent voltage [22] and equivalent flux vectors [27,74]. In [22], the deadbeat-predictive torque control was developed considering both the torque ripple and THD minimization based on the two-step modulation method. However, the available resources of OW configuration were underutilized as the switching states corresponding to zero ZSV were selected to minimize the torque and flux ripple.

Compared to the MPTC, the MPCC scheme is simpler as the control objective is related to stator currents only, thus weighting factors can be avoided [76]. Many studies have been proposed for OW-PMSM drive systems employing MPCC [21,23,24,57–62]. Zhang et al. [57] have proposed an MPCC algorithm for the OW-PMSM system considering 19 candidate voltage vectors on the α-β plane without ZSC suppression. To suppress the ZSC and improve the steady-state performance, the voltage space vectors of the dual converter have been extended from a 2-D plane to a 3-D space, and the optimal voltage vector is selected depending on the reference ZSV [23,24]. Cheng et al. in [58] proposed a simplified 3-D MPCC control strategy for OW-PMSM with common DC bus topology. The presented strategy replaced the cost function related to stator currents in conventional MPCC with associated voltage vectors. To reduce the computational burden, candidate voltage vectors and reference voltage vectors were placed into a 3-D space, and by utilizing space geometry theory, the optimal voltage vector was selected instantaneously. In addition, a voltage vector tracking (VVT)-MPCC was also proposed to increase the reference vector tracking accuracy, thus improving the system performance effectively. However, the system complexity increases, and the switching frequency is also high. To reduce these issues, the space vectors were transformed from $\alpha\beta0$ to $abc$ frame in [59]. It allowed us to select the optimal VV efficiently and optimized the switch state transition by taking advantage of redundant switching states. Both the computation time and switching frequency were reduced. A simple MPCC-based strategy for a semi-controlled OW-PMSM system was developed in [61], where the steady-state system performance was improved by adjusting the uncontrollable side voltage vector. To further reduce the computational burden, a low complexity deadbeat-predictive control was proposed in [21], where the optimal voltage vector is selected based on offline analysis. Due to the high time-space of the proposed control scheme, an extended observer can be added to make the system more robust against parameter mismatch. Most of the model predictive control algorithms discussed above do not include any disturbance observer. However, the machine parameter may change in real-time, and the performance of the proposed methods may suffer considerably. To improve the system performance against parameter variation, Yuan et al. [60] proposed a novel MPCC scheme for torque ripple suppression of OW-PMSM. Here, an adaptive sliding mode disturbance observer that can simultaneously estimate the ZSC and third harmonic back emf was developed considering the dead-time effect. To further eliminate the parameter mismatch in the $d$-axis, $q$-axis, and zero-sequence loop, Li et al. [62] formulated a control scheme that combined the deadbeat-predictive current control (DPCC) with an extended state observer. The reference [77] proposed a robust DPCC for OW-PMSM (surface type), which eliminated the effect of parameter variation by estimating the machine inductance. To enhance the torque performance, an extra current was injected into the reference $q$-axis current in combination with DPCC in [78]. A model free of predictive control scheme was presented in [79], which employs a sliding mode observer to estimate the $dq0$-axis current and produce a compensation current for torque ripple minimization.

Since a large body of recent research is related to MPC-based control of the OW-PMSM, a summary table can help to compare these proposed techniques in terms of various performance indices. Table 1 presents this comparative summary. One key factor that can be noticed from Table 1 is that the THD of phase current improves when the number of applied VV is two or more at every control period. More than one VV reduces the error between the reference and the actual VV. The low THD in the current also indicates ZSC suppression.

**Table 1.** Comparison among different MPC algorithms for OW-PMSM; N/A means not applicable.

| Methods | Paper | Num. of Candidate VVs | Number of Applied VVs | Calc. Time (μs) | Current THD | Torque Ripple |
|---|---|---|---|---|---|---|
| MPCC | Conventional MPCC | 27 | 1 | Extremely high | high | high |
| | [58] | 8 | 1 | Relatively low | high | high |
| | | 8 | 3 | Moderate | low | low |
| | [23] | 5–9 | 1 | Relatively low | high | high |
| | | 5–9 | 3 | Moderate | low | low |
| | [21] | (Offline calculation) | 1 | Lowest | Similar to conventional FCS-MPCC | Similar to conventional FCS-MPCC |
| | [24] (Semi-controlled) | 3 or 5 (Reduced from 49 VVs) | 2 | Relatively low | low | N/A |
| | [80] | 12 | 3 | Moderate | high (11.31% at rated speed) | High (23.5% At rated speed) |
| MPTC | [22] | 6 | 3 | Moderate | low | low |
| | [81] | 6 or 7 | 1 | Moderate (60.6 μs) | high | N/A |

### 4.4. Fault-Tolerant Control

Since the fault tolerance is one of the key benefits of an OW-PMSM, several different fault-tolerant control schemes were also proposed in the literature. Different types of faults in the PMSM were analysed in [82,83]. The open-phase fault was the most common [84], and several fault-tolerant control schemes were proposed for this fault alone. The key objective of a fault-tolerant scheme is to keep the torque ripple within the acceptable limit.

A simplified modulation strategy with conventional coordinate transformation has been proposed in [85] to improve the control performance for open-phase faults. Although the ripple content of *dq*-axis currents under open-phase fault has been suppressed, there is still substantial torque ripple. Moreover, the torque ripple suppression cannot be guaranteed in this case as the control scheme still adopts the conventional coordinate transformation after the open-phase fault, which makes the torque expression dependent on an electrical angle. Furthermore, the coupling terms in the *dq*-axis voltage equations affect the overall system performance. Chen et al. [86] introduced a new coordinate transformation to suppress the torque ripple by modifying the q-axis reference current for OW-PMSM with a common DC bus under open-phase fault. Although the torque ripple was effectively suppressed, the THD content of the phase current increased to 14.31%. By combining a predictive control strategy with an embedded vector resonant controller, Song et al. [87] proposed a ZSC suppression strategy for the OW-PMSM system under open-phase fault without reconfiguring the controller after the fault. Researchers in [31] utilized the zero-sequence current for the post-fault operation of the OW-PMSM system for a maximum of six diode/switch open-circuit faults. In [88], a leg-sharing-based fault-tolerant control strategy for a maximum of two-leg open-circuit fault was developed, in which asymmetric zero-sequence voltage was injected to suppress the torque ripple. To further enhance the fault-tolerant capability of OW-PMSM with common DC bus configuration for multiple open-leg faults, a winding reconnection-based strategy utilizing bidirectional solid-state relays has been presented in [32]. Here, the fault-tolerant operation for open-circuit fault up to three lags was possible with constant output capacity by keeping the three-phase voltages practically unchanged. The article in [31,32,85–87] discussed the post-fault control strategy for one fault only. These strategies cannot operate if a second fault occurs during

fault-tolerant control. To further improve the system reliability, the reference [89] presented a second-time fault-tolerant control strategy, based on the idea of bridge arm sharing, for OW-PMSM considering second-time open-circuit fault of the remaining semiconductor switches during one-phase open-circuit fault.

Most of the fault-tolerant control schemes only consider stator faults. The fault-tolerant techniques for rotor faults such as demagnetization of the rotor pole and eccentricity in the rotor shaft are yet to be evolved for OW-PMSM.

## 5. Performance Comparison among Different Methods

It can be noted from Section 4, many different control schemes have been proposed and investigated for the OW-PMSM. The existing control schemes are summarized in Table 2 to highlight the performance improvement achieved with these techniques. It can be observed from Table 2 that the THD of current and torque ripple vary widely under these schemes. Both current THD and torque ripple were minimum with FOC-based SVPWM compared to MPC and DTC, indicating more research is still required to bring the performance of MPC and DTC in parallel with FOC-based SVPWM. All existing control schemes rely heavily on machine parameters, and performance can suffer if the parameter uncertainty becomes significant. Use of an observer to make the control schemes of OW-PMSM more robust against the machine parameter variation is gaining attention in recent times [21,62,79]. One of the benefits of PI/PR-based vector controls is that it allows fixed switching frequency operation, but it suffers from slower dynamics compared to the DTC and MPC. In a WEC, low THD and torque ripple are more important than slow dynamics; hence, SVPWM-based techniques dominate generator control in WEC. However, non-linear control schemes such as MPC offers greater flexibility to incorporate more than one control objective. Hence, the OW-PMSM in WEC may benefit from evolving MPC techniques.

**Table 2.** Summary of the studied control schemes; N/A refers to not applicable.

| Methods | Swit. Freq. | Dyna-Mics | Param. Sensitivity | Calc. Time | SVPWM Modulator | Control Param. Tuning | Multi-Step Optimiza-tion | Paper | Current THD (Steady State) | Torque Ripple (Steady State) |
|---|---|---|---|---|---|---|---|---|---|---|
| SVPWM control | Fixed | Slower | Less sensitive | low | Required | PI and PR coefficient | Not supported | [10] | 4.39% | 1.79% |
| | | | | | | | | [14] | 7.75% | 10% |
| | | | | | | | | [11] | 7.56% | N/A |
| | | | | | | | | [54] | 54.8% | 1.15% |
| | | | | | | | | [66] | 3.08% | N/A |
| MPC | Variable | Faster | Sensitive | high | Not required | Not required for MPCC, required for MPTC if the cost function is related to both torque and flux | Supported | [60] | N/A | 2.5% |
| | | | | | | | | [58] | 8.06% | 10% |
| | | | | | | | | [24] (Semi-Controlled) | 8.33% | N/A |
| | | | | | | | | [81] | 30.31% | N/A |
| | | | | | | | | [80] | 11.31% | 23.5% |
| | | | | | | | | [61] (Semi-Controlled) | 9.28% | N/A |
| | | | | | | | | [22] | 6.83% | 7.2% |
| | | | | | | | | [23] | 13.38% | N/A |
| DTC | Variable | Faster | Sensitive | low | May/may not require | Not required | Not supported | [19] | 7.04% | 5.0% |
| | | | | | | | | [20] | 16.3% | 14% |

## 6. Challenges and Future Direction for the OW-PMSM

The review of various existing literature revealed that in terms of fault-tolerance capability, DC bus utilization, and overall size, the common DC bus configuration of OW-PMSM is superior compared to the other two topologies. However, the output power quality degrades for this configuration due to the presence of the zero-sequence current.

Using appropriate control schemes, the THD of the phase current and torque ripple can be significantly improved under both healthy and faulty conditions. Many of the existing control schemes of OW-PMSM have been investigated for motoring operations. It can be safely assumed that performance improvement brought by most of these techniques is equally valid for the generating operation.

Usually, wind turbines are located offshore or in mountainous areas, where access to maintenance work may be difficult. Hence, the generator drive system with fault tolerance capability is highly beneficial for such systems. Moreover, the direct-drive generators used in the WEC are low-speed machines with many magnet poles. The partial or full demagnetization of magnet poles is not rare. It will be economical if the generator can still be operated under partial demagnetization. The fault-tolerant control of OW-PMSM may offer such opportunities. The existing literature on fault-tolerant control of OW-PMSM mainly concentrates on open-phase faults [31,32,85–87]. The fault-tolerant control schemes for other faults, especially rotor faults such as demagnetization, are yet to be evolved.

Among the various control techniques, although MPC offers several benefits such as the multi-objective cost function optimization, the THD of phase current and torque ripple are relatively high for a grid-connected generator system, as the grid imposes stringent requirements on current THD (5% [90]). Performance deterioration due to parameter variation is also a yet-to-be-resolved issue. The use of observers such as a sliding-mode observer and a Luenberger observer to make the controllers robust against parameter variation is being explored. However, the increase in computational burden, especially in MPC, can make the adoption of observer techniques more challenging.

In recent times, many other advanced control schemes such as fuzzy logic controllers [91–93], artificial neural networks [94], and finite-time control [95,96] have been investigated for conventional PMSM. Adaptation of these controllers for the OW-PMSM will require further investigation.

## 7. Conclusions

A comprehensive review of the existing control algorithms for OW-PMSM with common DC topology has been presented in this article. The open-winding configuration offers several benefits over the conventional Y-connected PMSM such as enhanced fault tolerance capability, multi-level operation, more control flexibility due to a higher number of voltage vectors, and reduced converter capacity. It has been suggested that the OW-PMSM with common DC topology may be a good choice for wind-power applications in terms of simplified configuration, low cost, fault-tolerant capability, DC bus voltage utilization, and power quality if an appropriate control algorithm is in place. Various control strategies have been proposed for OW-PMSM with a common DC bus under healthy and open-circuit faults. However, the control algorithm for OW-PMSM under magnet fault conditions is yet to be developed. This might be an interesting area for future researchers, as the direct-drive wind turbine generator usually contains a large number of magnets, and some of these magnets can be demagnetized due to temperature, vibration, material corrosion, and high instant armature field. Moreover, it can be observed from the review that compared to the conventional PI/PR-based vector control algorithm, model-predictive control offers a fast dynamic response and simpler control structure and can handle multivariable and nonlinear systems. However, the THD of phase current and torque ripple was high for model-predictive control, and the performance of MPC degrades further due to the parameter mismatch. The high ripple content in stator current and torque may be attributed to the finite number of voltage vectors, as there will always be an error between the reference voltage vector (VV) and the applied actual VV. Therefore, the performance of MPC needs to be further enhanced to deploy this control scheme for open-winding wind turbine generators, as the grid requires the current THD to be within 5%. In addition, some advanced control strategies such as neural networks, fuzzy logic controllers, and finite-time control, which have already been implemented for Y-connected PMSM, can be further investigated for OW-PMSM to improve the steady-state and dynamic performance.

**Author Contributions:** Conceptualization, A.R. and R.D.; methodology, R.D.; software, A.R.; formal analysis, A.R.; resources, A.R., R.D. and G.C.; writing—original draft preparation, A.R.; writing—review and editing, R.D., G.C., M.F.R., D.X. and V.K.T.; visualization, R.D.; supervision, R.D., G.C., M.F.R. and D.X.; project administration, R.D. All authors have read and agreed to the published version of the manuscript.

**Funding:** This research received no external funding.

**Data Availability Statement:** No new data were created.

**Conflicts of Interest:** The authors declare no conflict of interest.

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
