# Peer review of "Open-Winding Permanent Magnet Synchronous Generator for Renewable Energy—A Review"

_energies, doi:10.3390/en16145268_

Round 1

Reviewer 1 Report

This summarize open-winding permanent magnet synchronous generator for renewable energy,

there are some questions that need to be answered, as follows:

(1) The importance and significance of this article should be emphasized in the introduction section;

(2) The references are too old, and more references in recent years should be given;

(3) Many advanced control methods have been proposed like sliding mode control (antisaturation adaptive fixed-time sliding mode controller design to achieve faster convergence rate and its application), finite-time control (antisaturation finite-time attitude tracking control based observer for a quadrotor), and neural networks (anti-saturation adaptive finite-time neural network based fault-tolerant tracking control for a quadrotor uav with external disturbances), observer technique (fixed-time disturbance observer-based robust fault-tolerant tracking control for uncertain quadrotor UAV subject to input delay), and the author can simply describe the suggested control methods so that the reader can learn more control methods;

(4) About further directions, the authors should summarize in more detail, which is also conducive to the scholars to have a deeper understanding of the future research work;

(5) The summary chapter lacks consideration for the future.

Minor editing of English language required.

Reviewer 2 Report

In this paper the authors present a review of control algorithms for OW-PMSG with DC topology.

In general, the paper is well written and well detailed. The body of work is well organized, and the way of presentation helps the reader. The bibliographic reference is adequate.

Considering the contribution of the paper as a reference source, it can be considered for publication after some considerations.

1- Despite the large amount of relevant information in the body of the paper, the conclusion presented is extremely succinct. I recommend that the authors rewrite the conclusions using results presented in the body of the paper, to validate the statements presented in the conclusion. For the reader, a well-presented conclusion serves as a starting point for reading.

The points below the conclusion should be better detailed using the results and references cited in the body of the paper:

a) The best configuration suitable for wind power generation application is OW-PMSG.

b) Why the authors consider that there is a gap in the research of wind systems, for CC bus?;

c) Why are predictive controls more indicated? What about the other non-linear controls? Why only consider the conventional PI/PR based vector control.
